# Genetic scores for predicting longevity in the Croatian oldest-old population

**Maja Šetinc**[1]*, **Željka Celinšćak**[1], **Luka Bočkor**[1,2], **Tanja Ćorić**[3], **Branko Kolarić**[3], **Anita Stojanović Marković**[1], **Matea Zajc Petranović**[1], **Marijana Peričić Salihović**[1], **Nina Smolej Narančić**[1], **Tatjana Škarić-Jurić**[1]

**1** Institute for Anthropological Research, Zagreb, Croatia, **2** Centre for Applied Bioanthropology, Institute for Anthropological Research, Zagreb, Croatia, **3** Andrija Štampar Teaching Institute of Public Health, Zagreb, Croatia

\* maja.setinc@inantro.hr

**Citation:** Šetinc M, Celinšćak Ž, Bočkor L, Ćorić T, Kolarić B, Stojanović Marković A, et al. (2023) Genetic scores for predicting longevity in the Croatian oldest-old population. PLoS ONE 18(2): e0279971. https://doi.org/10.1371/journal.pone.0279971

**Data Availability Statement:** Fully anonymised dataset used in this study is now publicly available on Zenodo repository (https://zenodo.org/record/7421684, DOI: 10.5281/zenodo.7421684).

## Abstract

Longevity is a hallmark of successful ageing and a complex trait with a significant genetic component. In this study, 43 single nucleotide polymorphisms (SNPs) were chosen from the literature and genotyped in a Croatian oldest-old sample (85+ years, sample size (N) = 314), in order to determine whether any of these SNPs have a significant effect on reaching the age thresholds for longevity (90+ years, N = 212) and extreme longevity (95+ years, N = 84). The best models were selected for both survival ages using multivariate logistic regression. In the model for reaching age 90, nine SNPs explained 20% of variance for survival to that age, while the 95-year model included five SNPs accounting for 9.3% of variance. The two SNPs that showed the most significant association ($p \leq 0.01$) with longevity were *TERC* rs16847897 and *GHRHR* rs2267723. Unweighted and weighted Genetic Longevity Scores (uGLS and wGLS) were calculated and their predictive power was tested. All four scores showed significant correlation with age at death ($p \leq 0.01$). They also passed the ROC curve test with at least 50% predictive ability, but wGLS90 stood out as the most accurate score, with a 69% chance of accurately predicting survival to the age of 90.

## Introduction

Continuous progress in reducing death rates during the early and middle years of life and improvements of the living conditions have resulted in a doubling of global life expectancy over the last two centuries [1], and according to data from the World Health Organization, that trend continues today [2] (accessed on 26th August 2022). This increase in life expectancy has led to a large increase in the percentage of older individuals in the population, and global predictions suggest that by 2050, for the first time in human history, there will be more people over 60 than adolescents and young adults combined. As old age is one of the main risk factors for the development of chronic illnesses such as cancer, cardiovascular and neurodegenerative diseases [3], the ageing of the population represents a significant burden on the social and healthcare systems of many countries [4]. Multimorbidity and frailty are also more prevalent among the elderly population [5, 6], often causing the need for long-term care in the later

**Funding:** This research was funded by Croatian Science Foundation grants IP-01-2018-2497 (HECUBA project) and DOK-2018-09-8382 to Tatjana Škarić-Jurić. The funders had no role in study design, data collection and analysis, decision to publish, or preparation of the manuscript.

**Competing interests:** The authors have declared that no competing interests exist.

stages of a person's life. This demographic phenomenon has brought to attention the importance of preventing age-related diseases and conditions, identifying phenotypes associated with healthy ageing and genetic variants and biomarkers underlying these traits [7], implementing a sustainable healthcare [8] as well as developing strategies to promote successful ageing. Longevity and healthy ageing, and how to achieve them, are therefore among the principal challenges in human biology and medicine today, and the importance of research on this topic will increase even more in the coming decades.

The first major breakthrough in ageing research was the discovery that caloric restriction could positively affect the lifespan of model organisms [9]. This finding has intrigued scientists for almost a century, and was tested and reproduced in other species as well, with the results from primates being published only recently [10, 11]. In recent years, caloric restriction has also been proposed as an approach to cancer prevention [12, 13] and disease management [14, 15]. Along with increasing lifespan, dietary restriction reduced the occurrence of age-related diseases [16], thus proving that it can also be beneficial for extending the healthspan—a term that refers to the total duration of life spent in good health and the part of the total lifespan free from illness. As human life expectancy continues to increase, the challenges of extending the healthspan become even more important [17, 18] in order to achieve "optimal longevity"–a long and high-quality life [18].

Another major finding that propelled ageing research even further was the discovery of a single gene, aptly named age-1, that affects the lifespan of *Caenorhabditis elegans* [19]. This discovery marked the beginning of a new era of genetics-based longevity research, in which conserved genes and interacting signalling pathways that contribute to longevity have been identified. The ageing process and age-associated phenotypes are linked via gene regulation [20], and the complex network of cellular pathways involved in this regulation has pointed to a much greater plasticity of the ageing process than previously believed [21]. Genetic studies conducted in recent years indicate the same conserved pathways discovered in model organisms may modulate lifespan and healthspan in humans as well [22]. These studies used a candidate gene approach to investigate their association with longevity. Genome-wide association studies (GWAS) are another type of study commonly used to uncover longevity loci in humans [23]. In order to reveal genetic variants that might contribute to reaching an advanced old age, the frequencies of genetic variants are usually compared between an aged group of interest and a younger control group. Studying such longevity loci could prove instrumental for determining the molecular mechanisms underlying healthy ageing, and could also enable accurate prediction of a person's chance of reaching old age.

Polygenic risk score is a sum of an individual's genetic risk for a disease or trait, and it could be a compelling tool for health and lifestyle management [24, 25]. While polygenic risk scores are usually constructed as linear combinations of individual variant effects [26], summing all risk variants reported for a disease on a genome-wide level, a genetic score for predicting the chance of survival to a threshold age of longevity is a sum of significant longevity loci. Genotype data for 43 SNPs previously associated with longevity were obtained for a sample of Croatian elderly individuals (85+ years of age), and this study explores the relation of these longevity variants with the age at death of the studied sample. Its main goals are:

- to find the most influential genetic variants in the Croatian oldest-old sample that are significantly related to longevity (90+) and extreme longevity (95+ years)

- to construct unweighted and weighted genetic scores and test their specificity and sensitivity to predict a chance of survival to the age of 90 and 95 years.

## Materials & methods

The study sample comprised 327 unrelated elderly individuals of both sexes aged 85 years and older, residents of the homes for elderly and infirm in Zagreb, the capital of Croatia (detailed description of the sample and study protocol could be found in Perinić Lewis et al. [27]). The informed consent was obtained from each study participant and the research was approved by the Ethics Committee of the Institute for Anthropological Research, Zagreb. The field study was conducted between 2007 and 2009, and 10 years after the initial sampling, the age at the time of death of each respondent was determined from the national mortality register.

43 single nucleotide polymorphisms (SNPs) in candidate genes for longevity were selected from publicly available literature databases (PubMed and repositories specialized for human longevity such as https://genomics.senescence.info/longevity/, http://ageing-map.org/). The SNPs were selected based on their strong or repeatedly reported association with human longevity and involvement in various metabolic pathways. S1 Table contains information about the selected SNPs (rs code, chromosome position, nearest gene, allele frequencies, MAF, genotyping success rate, HWE p-values and information on literature mentioning association with longevity).

Each participant provided a peripheral blood sample, and genomic DNA was isolated from leukocytes using the salting-out method [28]. Genotyping was outsourced and done in a commercial laboratory using a Kompetitive Allele Specific PCR (KASP) [29]. It is a genotyping assay that combines competitive allele-specific PCR with a homogeneous fluorescence-based reporting system for the identification and measurement of genetic variation occurring at the nucleotide level to detect SNPs or insertions and deletions (InDels). After genotyping, the final sample comprised 314 participants, as 13 participants had missing data on nine or more SNPs (>20% of unsuccessfully genotyped loci) and were therefore excluded from the analysis. Exclusion criteria were determined according to the principle of parsimony to retain the highest possible number of participants. Because all genetic data for each participant needed to be complete to calculate a genetic score, all missing data for participants with 1–8 unsuccessfully genotyped SNPs were replaced by the median value for that SNP.

Genotype data (available in open access on the online repository Zenodo [30]) were coded for each participant as follows: a value of 2 was assigned to the homozygous genotype of longevity allele, a value of 1 to the heterozygous genotype, and a value of 0 to the homozygous genotype of an allele not associated with longevity in our sample. In cases where less than 10 homozygous genotypes of any type were determined, and in cases of SNPs where dominant or recessive coding proved more significant in further analyses (rs2267723, rs16847897), they were coded with only the values 0 and 1, and the heterozygote was added to the less common homozygote. The coded data were used to perform univariate logistic regression as a means of selecting the SNPs that have a potential influence on longevity, using a cut-off p-value $\leq 0.20$ [31]. Two separate analyses were performed with age at death as the dependent variable, for both of which the participants were divided in two groups—a group of those who died before, and a group of those who died after reaching a cut-off age of 90 or 95 years. The number of participants in the two groups according to their age at death was: for the cut-off age 90, there were 103 participants who died before the age of 90, and 211 participants who lived over 90; for the cut-off age 95, there were 230 participants who died before reaching 95 years of age, and 84 participants who lived over 95. All SNPs that had a p-value $\leq 0.20$ in univariate analysis were selected for testing in a multivariate logistic regression model. The best models for age thresholds of 90 and 95 years were selected for further calculations.

Genetic Risk Score is called Genetic Longevity Score (GLS) in this study, since "risk" for reaching the age of 90 or 95 is a preferred trait, and thus a more appropriate term was chosen. GLS is a number representing a sum of alleles associated with human longevity across loci

included in the best multivariate logistic model. Unweighted GLS (uGLS) was calculated by summing the coding values assigned to genotypes at all SNPs that accounted for the best logistic regression model for ages at death 90 or 95 (uGLS90 and uGLS95, respectively). Weighted GLS (wGLS90 and wGLS95) was calculated by summing the genotype values for each SNP multiplied by their respective exponentiation of the beta coefficient from the multivariate model. To test the reliability of the scores, additional statistical analyses were performed to evaluate their association with age at death as a continuous or discrete variable (i.e. descriptive analysis, Pearson's correlations, Chi-square test, multiple regression analyses). Receiver operating characteristics (ROC) curve analysis was performed for both unweighted and weighted GLS, with binary age at death set as the dependent variable to calculate the area under the curve (AUC). All statistical calculations were performed using the SPSS software package 21.0.

## Results

The general information on investigated 43 SNPs is presented in S1 Table, and univariate logistic regression results for all 43 SNPs with survival ages of 90+ and 95+ years as the dependent variable are shown in S2 Table. In univariate analysis, five SNPs (rs3772190, rs16847897, rs1800629, rs2267723, rs7412) were significantly associated (p-value of $\leq 0.05$) with survival to the age of 90, and one SNP (rs429358) to survival to the age of 95. Since only one SNP was entered in the logistic regression in this analysis, this p-value did not have to undergo multiple correction testing. This shows a strong correlation between these SNPs with longevity in the studied population. However, in order to enlarge the qualifying pool of SNPs for further analyses, p-value of $\leq 0.2$ was selected as the cut-off value for SNPs to be entered into multivariate logistic regression analysis [31]. With this inclusion criteria for the multivariate analyses, 17 SNPs entered the series of models for survival age 90+, and 10 SNPs entered the models for age 95+. The best multivariate models, which explain the largest proportion of variance in survival age, are presented in Tables 1 and 2, as well as in Fig 1, which shows a forest plot of SNPs that are positively (OR > 1) associated with longevity.

The best model, explaining 20.5% of the variance in survival to 90+ years of the oldest-old Croatian sample, has nine SNPs and is presented in Table 1. The two SNPs that showed the most significant association (p $\leq 0.01$) are: rs16847897, located in the *TERC* gene with the more frequent homozygote (GG) having a 2.128 times higher chance (95% CI 1.249–3.627, p = 0.005), and rs2267723 in the *GHRHR* gene, whose less common homozygote (AA) has a 2.280 times higher chance (95% CI 1.239–4.194, p = 0.008) of reaching 90 years of age. A lower degree of significance (p $\leq 0.05$) has the locus rs7412, in the *APOE* gene, where the carriers of the less frequent allele T (genotypes TT and CT) have a 3.055 times greater chance of living over 90 years (95% CI 1.230–7.587, p = 0.016) than the homozygotic carriers of allele C, and the locus rs1800629, upstream of the *TNF-α* gene, whose more common homozygotes (GG) are 1.898 times (95% CI 1.038–3.468, p = 0.037) more likely to survive up to the age of 90. Finally, rs1042522 located in the *TP53* gene did not reach statistical significance at the level of the entire locus, but heterozygotes for this locus (CG) have a 1.752 times (95% CI 1.010–3.040, p = 0.046) higher chance of reaching 90 years of age. The additional four loci—rs12206094 (in the *FOXO3* gene), rs9536314 (in the *KLOTHO* gene), rs50871 (in the *ERCC2* gene) and rs17202060 (in the *TXNRD1* gene)–are also included in the best model for the survival age of 90 years because they contribute to the quality of the model. Out of five SNPs that were significant at the univariate level, only one wasn't included in the best multivariate model. That SNP was rs3772190, located in the *TERC* gene, which was excluded due to its linkage (calculated in Haploview software [32]) with another *TERC* SNP, rs16847897, which entered the multivariate model since it showed a stronger association with survival to the age of 90.

**Table 1. The best multivariate logistic regression model for survival to the age of 90 years in the Croatian oldest-old sample (N = 314).**

| Closest gene | SNP | Contrasting genotypes | B | p | O.R. | 95% C.I. for O.R. | |
|---|---|---|---|---|---|---|---|
| | | | | | | Lower | Upper |
| *APOE* | rs7412 | CC vs TT, CT | 1.117 | **0.016** | 3.055 | 1.230 | 7.587 |
| *ERCC2* | rs50871 | CC vs AC vs AA | | 0.238 | | | |
| | | CC, AA vs AC | -0.407 | 0.229 | 0.665 | 0.343 | 1.292 |
| | | CC, AC vs AA | 0.069 | 0.856 | 1.072 | 0.506 | 2.268 |
| *FOXO3* | rs12206094 | CC vs TC vs TT | | 0.092 | | | |
| | | CC, TT vs TC | -0.363 | 0.183 | 0.696 | 0.408 | 1.187 |
| | | CC, TC vs TT | 0.847 | 0.159 | 2.332 | 0.717 | 7.583 |
| *GHRHR* | rs2267723 | GG, AG vs AA | 0.824 | **0.008** | 2.280 | 1.239 | 4.194 |
| *KL (KLOTHO)* | rs9536314 | TT vs GG, TG | 0.454 | 0.181 | 1.575 | 0.809 | 3.065 |
| *TERC* | rs16847897 | CC, GC vs GG | 0.755 | **0.005** | 2.128 | 1.249 | 3.627 |
| *TNF-α* | rs1800629 | AA, GA vs GG | 0.641 | **0.037** | 1.898 | 1.038 | 3.468 |
| *TP53* | rs1042522 | CC vs GG vs CG | | 0.092 | | | |
| | | CC, CG vs GG | 0.833 | 0.248 | 2.300 | 0.559 | 9.456 |
| | | CC, GG vs CG | 0.561 | **0.046** | 1.752 | 1.010 | 3.040 |
| *TXNRD1* | rs17202060 | TT vs CC vs TC | | 0.164 | | | |
| | | TT, TC vs CC | 0.292 | 0.480 | 1.339 | 0.595 | 3.012 |
| | | TT, CC vs TC | 0.705 | 0.094 | 2.024 | 0.887 | 4.621 |

| Nagelkerke R-squared | 0.205 |
|---|---|
| Hosmer—Lemeshow test | 0.536 |
| % Correct | 72.9 |

This table shows all the SNPs that together make up the best model for predicting survival to age 90+, the genotypes that were contrasted within the model, beta values, odd ratios and 95% confidence intervals. The p-values of SNPs that passed the significance threshold of $p \leq 0.05$ are highlighted in bold. Nagelkerke R-squared value, indicating the amount of variance explained by the model, is shown at the bottom of the table along with the results of Hosmer—Lemeshow test and the percentage of correctly classified cases.

Table 2 presents the best model for predicting survival to age 95+, explaining 9.3% of variance. Of the five SNPs contributing to the best model, no locus was significant at the entire locus level (three contrasting genotypes). However, there are several significant associations that elucidate specific genotypes: the rs6067484 locus in the *PTPN1* gene, whose less frequent homozygotes (GG) have a 2.505 times higher chance for reaching 95 years of age (95% CI 1.049–5.981, p = 0.039). Also, for rs4837525 located in the *PAPPA* gene, heterozygotes (AG) have a chance of living over the age of 95, which is 2.703 times (95% CI 1.039–7.033, p = 0.042) higher than those of both homozygotes, and for rs1042522 of the *TP53* gene, the less common homozygote (GG) has a 3.233 times (95% CI 1.013–10.322, p = 0.048) higher chance of surviving to the age of 95 years. Finally, rs429358 in the *APOE* gene, which was also significant on the univariate analysis level, is associated with survival to 95 years at the $p \leq 0.1$ significance level (p = 0.053), with the more frequent genotype (TT) having a 2.345 times (95% CI 0.988–5.563) greater chance of reaching 95 years of age. The association of rs12203592 in the *IRF4* gene, although not statistically significant, contributes to the strength of the model.

It should be noted that the two selected models share only one locus—rs1042522 in the *TP53* gene—which is significantly associated with survival to age 90+ (CG genotype) as well as to age 95+ (GG genotype). However, both models also indicate epsilon diplotypes of the *APOE* gene: the first model at the rs7412 locus and the second at the rs429358 locus. Data on the frequencies of *APOE* gene longevity loci and APOE isoforms in the oldest-old Croatian population are shown in Table 3, with allele distribution frequencies in the European population

**Table 2. The best multivariate logistic regression model for survival to the age of 95.0 years in the Croatian oldest-old sample (N = 314).**

| Closest gene | SNP | Contrasting genotypes | B | p | O.R. | 95% C.I. for O.R. | |
|---|---|---|---|---|---|---|---|
| | | | | | | Lower | Upper |
| APOE | rs429358 | CC, CT vs TT | 0.852 | 0.053 | 2.345 | 0.988 | 5.563 |
| IRF4 | rs12203592 | CC vs CT, TT | 0.569 | 0.110 | 1.766 | 0.880 | 3.546 |
| PAPPA | rs4837525 | AA vs AG vs GG | | 0.119 | | | |
| | | AA, AG vs GG | 0.766 | 0.127 | 2.151 | 0.804 | 5.757 |
| | | AA, GG vs AG | 0.994 | **0.042** | 2.703 | 1.039 | 7.033 |
| PTPN1 | rs6067484 | AA vs GA vs GG | | 0.116 | | | |
| | | AA, GG vs GA | 0.116 | 0.685 | 1.123 | 0.640 | 1.970 |
| | | AA, GA vs GG | 0.918 | **0.039** | 2.505 | 1.049 | 5.981 |
| TP53 | rs1042522 | CC vs CG vs GG | | 0.123 | | | |
| | | CC, CG vs GG | 1.174 | **0.048** | 3.233 | 1.013 | 10.322 |
| | | CC, GG vs CG | 0.245 | 0.375 | 1.278 | 0.744 | 2.197 |

| | |
|---|---|
| Nagelkerke R-squared | 0.093 |
| Hosmer—Lemeshow test | 0.763 |
| % Correct | 74.2 |

This table shows all the SNPs that together make up the best model for predicting survival to age 95+, the genotypes that were contrasted within the model, beta values, odd ratios and 95% confidence intervals. The p-values of SNPs that passed the significance threshold of p ≤ 0.05 are highlighted in bold. Nagelkerke R-squared value, indicating the amount of variance explained by the model, is shown at the bottom of the table along with the results of Hosmer—Lemeshow test and the percentage of correctly classified cases.

taken from the gnomAD database [33], and isoform frequencies of 1038 control subjects of European origin under 60 years of age, taken from the paper of McKay et al. [34].

Unweighted and weighted longevity scores were calculated for survival ages 90+ and 95+. Descriptive data of unweighted and weighted GLS90 and GLS95 are shown in Table 4, while the distribution of all four genetic longevity scores by the low/high division of the age-at-death variable is presented in Fig 2. The mean value of the scores within individual age-at-death groups is also presented. Pearson's correlation of all four longevity scores with age at death as a continuous variable is presented in Table 5. With p-value ≤ 0.01, all four GLSs were significantly associated with age at death. There was no significant difference in any calculated GLS between sexes, which is shown in S3 Table.

The distribution of the two unweighted genetic longevity scores (uGLS90 and uGLS95) in the three age-at-death groups (≤ 90.00 years, 90.00–94.99 years, and 95.00+ years) is presented in Fig 3. None of the highest-scoring participants for uGLS90 died before the age of 90, while the entire range of scores were represented in the lowest survival group for uGLS95. The lowest longevity scores were found among participants who died before the age of 90, and were not observed in the other two groups. The distribution is similar between the two scores among those who lived the longest, with no participant having a score below three. In the higher survival groups, distribution curves shift to the right side of the x-axis and higher longevity scores. S1 Fig shows the distribution of median values of genetic longevity scores by three age-at-death groups: <90.00, 90.0–94.99, and 95.00+, presenting the absolute number of individuals in each group. All scores yield analogous results: the group of participants who died before the age of 90 has a higher percentage of below-median longevity scores. Likewise, participants who survived beyond the age of 95 have a higher percentage of above-median scores. The relative age distribution of age-at-death groups (the percentage of each group is equal to 100%) according to the median of weighted genetic longevity scores is presented in S2 Fig, which

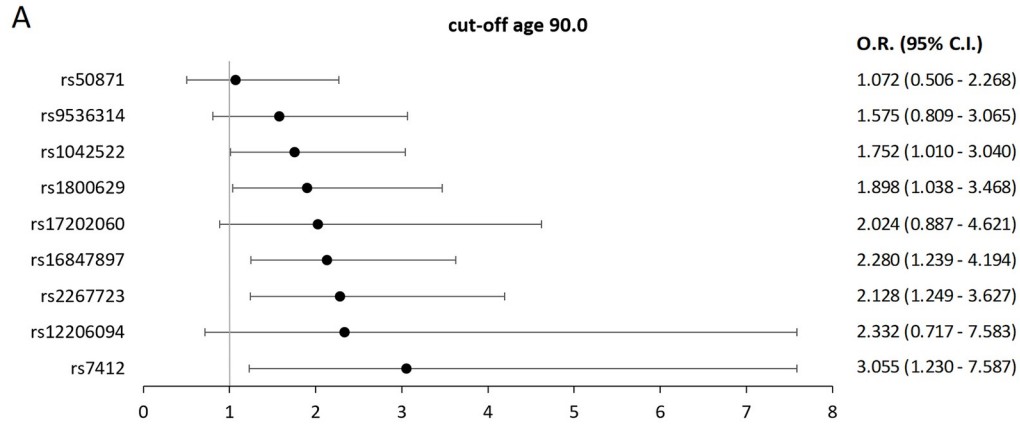

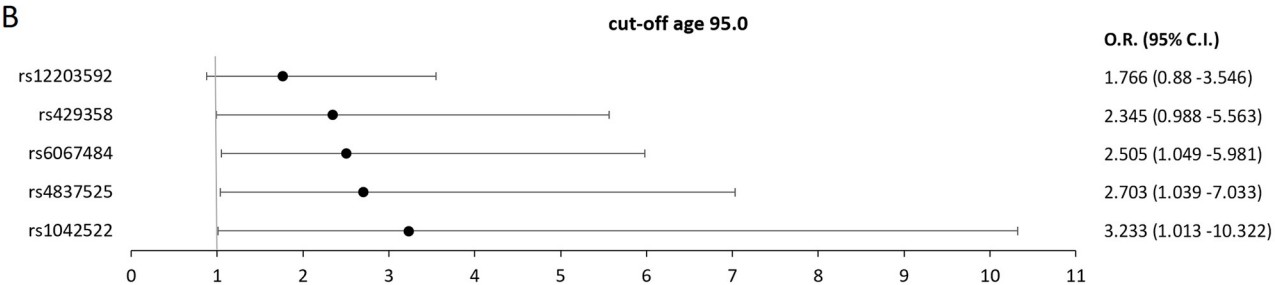

**Fig 1. Forest plot of SNPs positively (O.R. > 1) associated with longevity, with multivariate model Odds Ratios (O.R.) and 95% confidence intervals (C.I.) displayed.** A) SNPs positively associated with survival to the age of 90, B) SNPs positively associated with survival to the age of 95.

demonstrates that the weighted longevity score for the threshold of 95 years is gradual and inverse between groups below and above the median. The largest proportion of participants who died before age 90 is in the below-the-median group, and the largest proportion of those who survived over age 95 is in the above-the-median wGLS95 group. On the other hand, the weighted longevity score for the threshold age of 90 years has equal distributions between the 90.00–94.99 group and the 95.00+ group for both below- and above-the-median scores, while the proportion of people who died before 90.00 year of age is substantially higher in the below-the-median wGLS90 group.

**Table 3. Frequencies of alleles at loci that determine APOE isoforms in a Croatian sample of oldest-old people, and their frequencies in the general European population.**

| | | Croatian oldest-old sample | | European frequencies | |
|---|---|---|---|---|---|
| | | allele T | allele C | allele T | allele C |
| **APOE loci** | rs7412 | 0.066 | 0.934 | 0.077 | 0.923 |
| | rs429358 | 0.914 | 0.086 | 0.851 | 0.149 |
| **APOE isoform frequencies** | ε2 (rs7412-T, rs429358-T) | | 0.076 | | 0.091 |
| | ε3 (rs7412-C, rs429358-T) | | 0.844 | | 0.733 |
| | ε4 (rs7412-C, rs429358-C) | | 0.079 | | 0.176 |

Allele frequencies in European populations were taken from the gnomAD database [33], and isoform frequencies of 1,038 control subjects of European origin under the age of 60 from a paper by McKay et al. [34].

**Table 4. Descriptive statistics of unweighted (uGLS) and weighted genetic longevity scores (wGLS) for survival to ages 90 and 95.**

|  | TheoreticalMaximum | Minimum | Maximum | Range | Mean | Std. Deviation |
|---|---|---|---|---|---|---|
| **uGLS90** | 18.000 | 1.000 | 15.000 | 14.000 | 7.869 | 2.551 |
| **wGLS90** | 36.232 | 1.330 | 29.271 | 27.941 | 14.336 | 5.132 |
| **uGLS95** | 10.000 | 1.000 | 9.000 | 8.000 | 4.379 | 1.510 |
| **wGLS95** | 25.104 | 2.703 | 22.599 | 19.896 | 11.112 | 3.732 |

In order to test the predictive power of the obtained genetic longevity scores, we performed ROC curve analysis (Fig 4), which showed that all four scores were satisfactory for predicting the possibility of reaching the longevity milestones (90+ and 95+ years) that were above the theoretical cut-off value for this test of 0.5 [35, 36]. However, with an AUC score of 0.690, wGLS90 is a more predictive longevity score for survival to the age of 90, and uGLS90 with an AUC score of 0.662 is the less predictive. ROC curve analysis showed no differences between

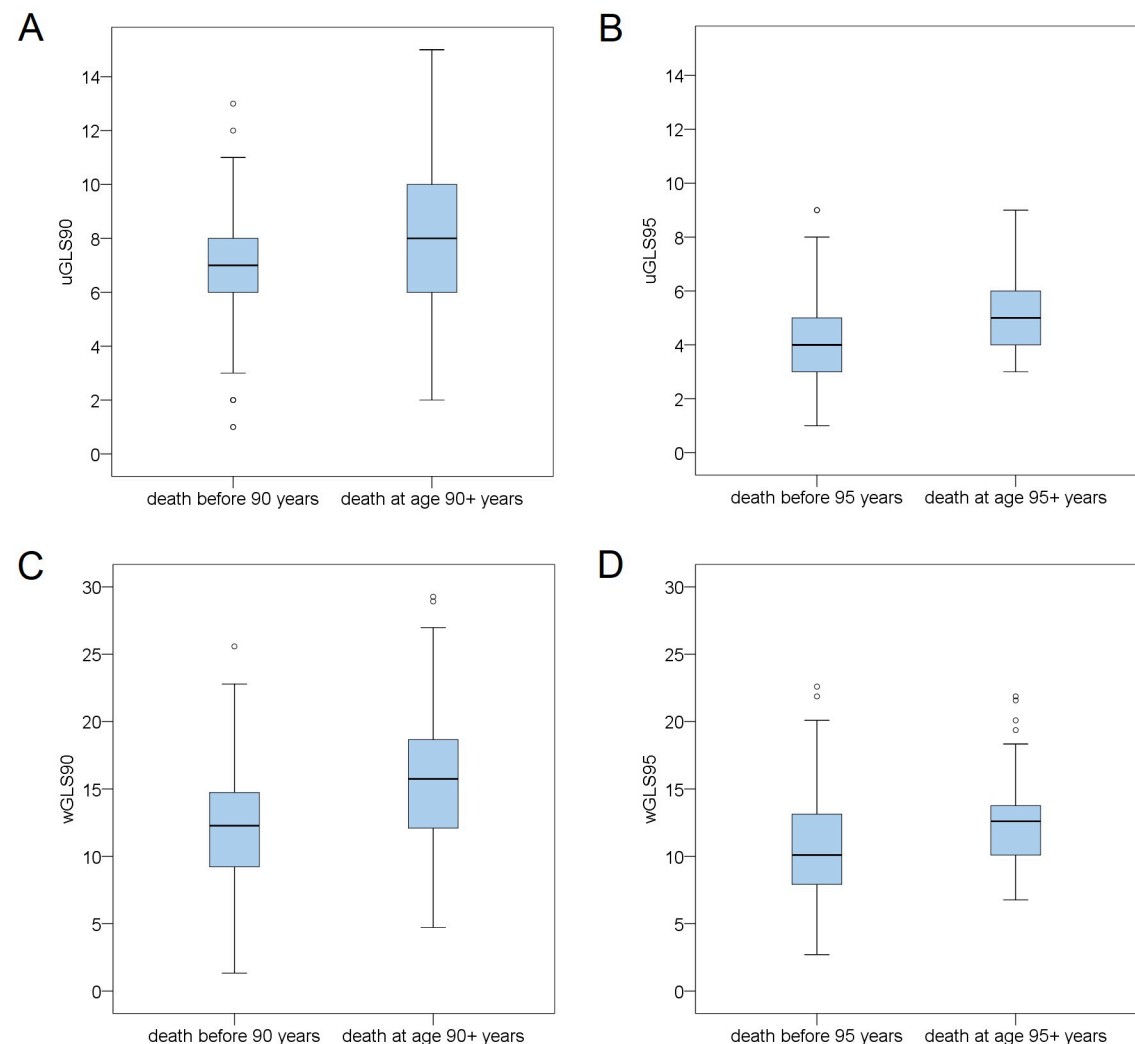

**Fig 2. Comparison of the genetic longevity score values between participants who died before and after reaching the cut-off ages of 90 and 95 years.** Box-and-whiskers plot showing the median value, quartile and extremes of A) uGLS90, B) uGLS95, C) wGLS90, D) wGLS95.

**Table 5. Correlation between age at death and calculated genetic longevity scores.**

|  | uGLS90 | wGLS90 | uGLS95 | wGLS95 |
|---|---|---|---|---|
| **Pearson correlation (r)** | 0.159 | 0.178 | 0.215 | 0.211 |
| **p** | 0.005 | 0.002 | 0.000 | 0.000 |

uGLS95 and wGLS95 as they are equally predictive with AUC scores of 0.649. A multivariate linear regression analysis was also performed, including all genetic scores as independent variables and a continuous age-at-death variable as a dependent phenotype. Beta-values and significance levels obtained from this analysis are presented in Table 6, with wGLS90 and uGLS95 highlighted as more informative scores.

## Discussion

With the continuing demographic trend of population ageing, achieving a long and healthy life is becoming more than a personal goal—it is now a research focus of scientists all around the globe. There is a great inter-individual variation in the rate at which one ages, and twin studies have shown that this variation, like longevity, has a genetic component [37, 38]. The principal aims of this study were to investigate which of the alleles associated with longevity in previous studies were important for reaching longevity thresholds in the Croatian oldest-old population (aged 85+), and to calculate and test unweighted and weighted genetic risk scores for predicting survival to the age of 90 and 95 using loci that proved significant in logistic regression analysis. The model for predicting survival to age 90 accounts for two times more variance than the model for predicting survival to age 95, which can be explained by fewer SNPs entering the 95-year model. We also suggest that a decline in the proportion of variance explaining the age at death is due to a shift between survivor and non-survivor groups and a decline in the expected outcome group. However, we cannot rule out the importance of some other stochastic elements that might also increase with advancing age, which reduces the genetic effect.

The two loci most strongly associated with reaching the age of 90 in the studied population are rs16847897 in the *TERC* gene and rs2267723 in the *GHRHR* gene. The *TERC* gene encodes for the RNA component of telomerase, a ribonucleoprotein that elongates telomeric DNA [39,

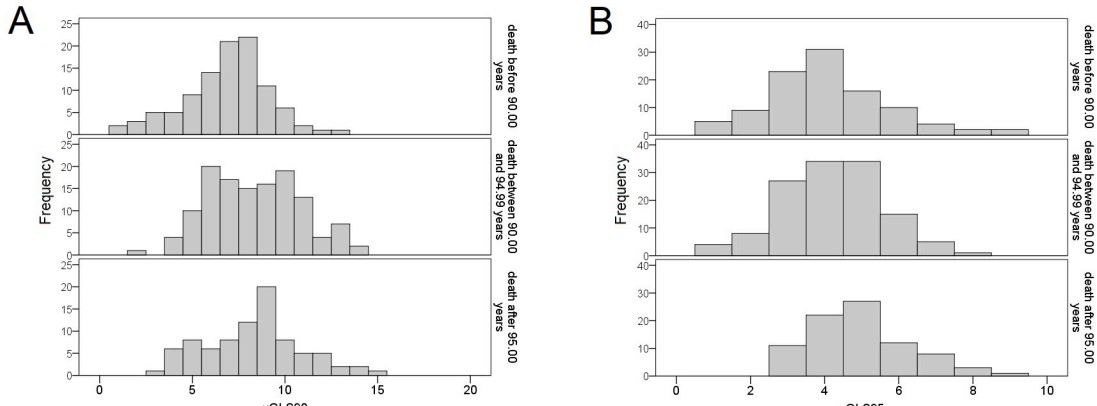

**Fig 3. Distribution of two unweighted genetic longevity scores in three age-at-death groups.** Histograms show the distribution of A) uGLS90, and B) uGLS95 among the participants belonging to a specific age-at-death group: < 90.00 years, 90.00–94.99 years, and 95.00+ years.

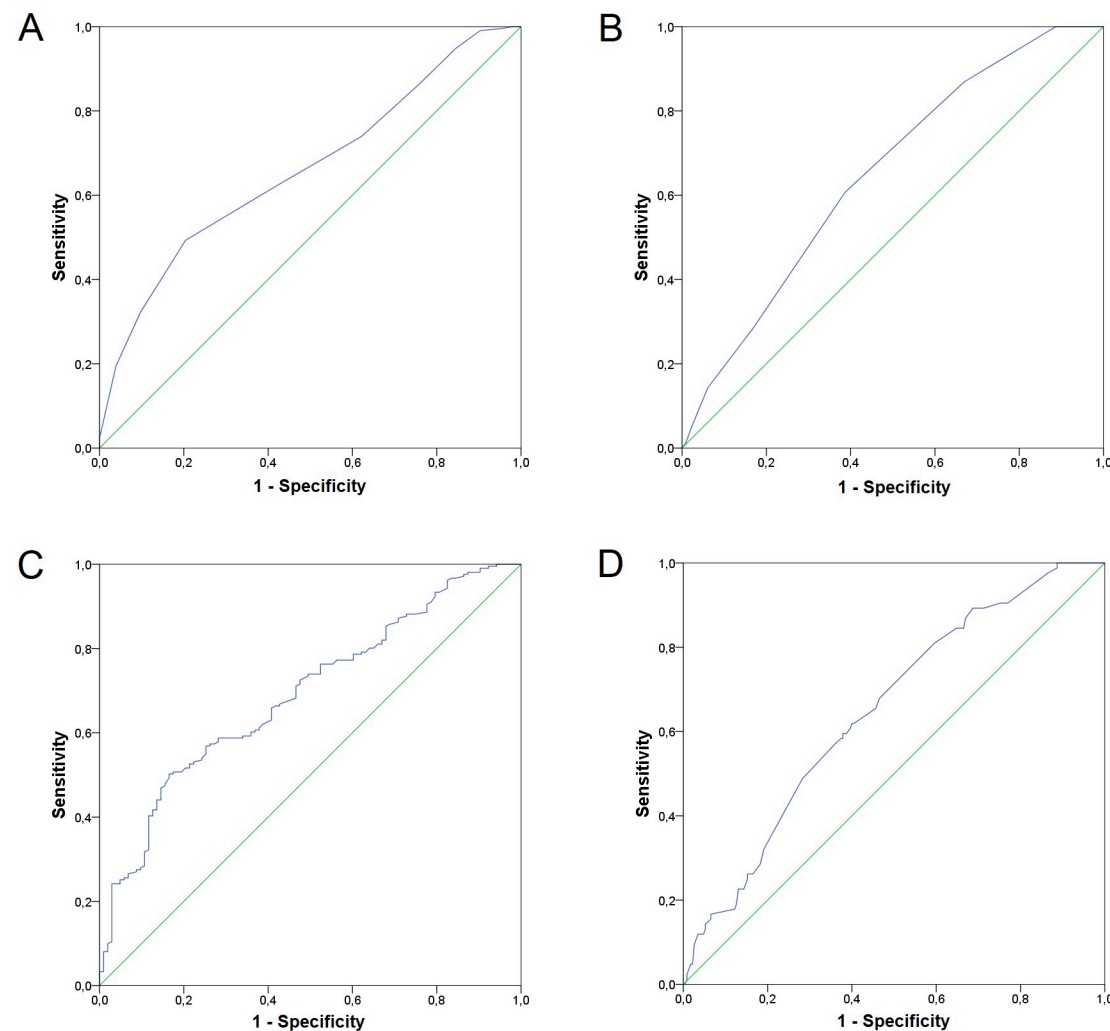

**Fig 4. Receiver operating characteristics (ROC) curves for calculated genetic longevity scores.** The ROC curve and area-under-curve (AUC) score for: A) uGLS90 (AUC = 0.662), B) uGLS95 (AUC = 0.649), C) wGLS90 (AUC = 0.690), D) wGLS95 (AUC = 0.649).

40]. Shorter leukocyte telomere length is frequently reported in patients who suffer from age-related diseases such as Alzheimer's disease [41] and vascular dementia [42], and has been proposed as a marker of biological ageing [43]. The intronic SNP rs16847897 located downstream of the *TERC* gene has been associated with leukocyte telomere length in large UK cohorts [44] and in the Chinese Han population [45]. The Chinese Han population study showed that the C allele of rs16847897 was associated with a shorter mean telomere length that equated to approximately 4 years of average age-related telomere attrition. Additionally, a study from

**Table 6. The results of a multivariate linear regression analysis including all genetic scores as independent variables and the continuous age-at-death variable as the dependent phenotype.**

|  | uGLS90 | wGLS90 | uGLS95 | wGLS95 |
|---|---|---|---|---|
| **Multiple regression (beta)** | -0.132 | 0.143 | 0.188 | -0.106 |
| **p** | 0.513 | 0.011 | 0.001 | 0.734 |

Scarabino et al. showed that the C allele increased the risk of earlier onset of Alzheimer's disease in the population from Southern Italy [46]. In the studied Croatian sample, the other allele, G, is beneficial for longevity because it contributes to the chances of reaching the longevity threshold of 90 years. In a follow-up study on a Southern Italian population, however, there was no significant association of rs16847897 with human lifespan [47]. Allele frequencies of this SNP vary considerably in different populations, so that the longevity allele G is a major allele in all mentioned European populations and a minor allele in the Chinese Han population.

The *GHRHR* gene encodes a growth hormone-releasing hormone receptor, a G protein-coupled receptor located on the membrane of somatotropic cells, cells that produce growth hormone in the anterior pituitary gland [48]. It binds growth hormone-releasing hormone, a peptide hormone produced in the hypothalamus. This binding is necessary for the proliferation of somatotrophs and for synthesis and secretion of growth hormone (GH) [49]. It is a part of the growth hormone/ insulin-like growth factor 1/insulin (GH/IGF-1/INS) signalling axis. Research from over 40 years ago showed that the secretion of GH and IGF-1 slowly decreases after an organism matures to adulthood, reaching its absolute lowest level in people over the age of 60 [50]. This biological phenomenon, which has been confirmed both in humans and other mammals [51], was even given a name–'somatopause' [52]. The decrease of GH/IGF-1 signaling has proven to extend longevity in many model organisms, including yeast, worms, fruit flies, and mice [53]. The minor allele A in the intronic rs2267723 of the *GHRHR* gene was significant for longevity in the Danish population [54]. It was also among the top-ranked interactions in a study that explored the combined effect of SNPs from candidate pathways on longevity [55]. This is in line with our study, where the minor allele A is advantageous for reaching 90 years of age.

The impact that changes in the immune system can have on ageing and reaching longevity has been clearly demonstrated in multiple studies [56]. With advancing age, the effectiveness of the immune response decreases, while inflammatory processes increase, which is described by the term 'inflamm-aging' [57]. This lack of equilibrium in the organism's response to stressors contributes to the development of chronic diseases with inflammatory pathogenesis, which are a major characteristic of ageing [58]. A significant association with survival to age 90 was found for rs1800629, located in a regulatory region upstream of the *TNF-α* gene. The *TNF-α* gene encodes a proinflammatory cytokine involved in many biological processes—from regulating proliferation, differentiation and apoptosis, to playing a role in lipid metabolism and coagulation. It has been linked to a number of conditions, including autoimmune disorders, insulin resistance, and cancer [59]. According to functional studies, the uncommon allele A of rs1800629 is a far more effective transcriptional activator than the common allele G [60]. However, a study conducted on an English Longitudinal study sample showed that the A allele is a risk factor for frailty [61], and similar results were obtained in a study of longevity and ageing of the Chinese population, where homozygous carriers of the A allele had worse results in physical function tests (Timed Up and Go Test and 5-meter walking test) [62]. In the Croatian oldest-old sample, the allele beneficial for reaching 90 years of age was the major allele G, which is in concordance with the previously mentioned research by Melki et al. and Yao et al. [61, 62].

rs1042522 was the only SNP significantly associated with survival to both ages 90 and 95. It is a missense variant with a very diverse distribution in world populations [63]. This SNP is located in the *TP53* gene, a gene that encodes the p53 protein that acts as a tumour suppressor by blocking cell cycle progression and promoting apoptosis. The p53 protein plays a central role in cellular regulatory pathways and is an important regulator of the expression and activity of several replication and transcription factors. Its activation is triggered by stress signals that

arise in response to the cell's conditions and environment. Some stressors, for example, are genotoxic damage, oncogene activation, replication stress, loss of normal cell connections and hypoxia [64]. It is crucial for determining cell fate by promoting either repair, survival, or elimination of damaged cells [65]. The *TP53* is the most frequently mutated gene in human cancer, and mutations in this gene can be found in >50% of all human cancers [66–68]. It is also of great importance for the ageing process, since apoptosis and cellular senescence strongly influence the homeostasis of tissues, and too much of both can deplete renewable tissues of progenitor or stem cells and reduce their ability for regeneration [69]. In the presence of intracellular reactive oxidative species (ROS), p53 becomes a target of the histone deacetylase SIRT1 [70, 71], whose expression is strongly down-regulated in senescent cells, and is often considered a potential target for longevity extension [70]. Furthermore, it can downregulate the insulin/IGF-1 pathway, which has been shown to increase lifespan [72]. The polymorphism of rs1042522 is a functional mutation that results in either an arginine (Arg) or a proline (Pro) residue at codon 72, with the proline allele showing a weaker response to induce apoptosis and prevent cell transformation [73, 74]. The European distribution of the Arg72Pro substitution is approximately 60%, 30% and 10% for Arg/Arg, Arg/Pro and Pro/Pro, respectively [75], with major C allele coding for arginine and minor G allele coding for proline. In a study by Ørsted et al. of the general Danish population, overall survival was higher for carriers of the G allele, both homozygotes (6% better survival) and heterozygotes (3% better survival), along with reduced mortality after cancer diagnosis [76]. Similar results were shown by a study conducted on a sample from the Leiden 85-plus study, in which the authors showed that carriers of the Pro/Pro genotype (G allele homozygotes) older than 85 years have increased survival compared to the carriers of Arg [77]. In a smaller cohort of 155 long-lived individuals, Groß et al. found that the proline allele was significantly associated with increased survival time in female participants [78]. In our study, rs1042522 was the only SNP significantly associated with both survival up to the age of 90 (CG genotype, Arg/Pro) and to the age of 95+ years (GG genotype, Pro/Pro). Therefore, regardless of the genotype, the G allele, which codes for proline at the 72nd residue of p53, proved to be beneficial for longevity in Croatian oldest-old persons, which coincides with the results of other studies. Interestingly, in the same paper on the Leiden 85-plus cohort, the Pro/Pro genotype was found as a risk for cancer mortality [77]. Furthermore, in a case-control association study of breast cancer performed on a sample of Croatian women, the percentage of Pro/Pro genotype was higher in cases (11.6%) than in controls (4.6%) [68]. Given the previously mentioned characteristic of the Pro allele for a reduced apoptotic response, that is perhaps not surprising. A reduced affinity for inducing apoptosis may cause a malignant cell being more likely to escape programmed cell death, thus increasing the risk of cancer. However, if Pro/Pro genotype triggers less apoptotic events, this might lead to a greater number of cells in general, which becomes increasingly important with old age. As an organism ages, proliferative capacity of tissues goes down, and this process might even be accelerated by an increased clearing of cells by apoptosis. Therefore, while the largest benefit for survival to the age of 90 in our studied population comes from the heterozygous Arg/Pro genotype (pointing to the importance of balance between cell proliferation and programmed cell death, and a possible heterosis effect), for survival to the threshold of extreme longevity (95 years) the maintenance of proliferative abilities that might come from homozygous Pro/Pro genotype seems to be more beneficial than cancer-protective effects of Arg. This could possibly explain the interplay through which p53 affects the ageing process and longevity. Nonetheless, we find it important to note that while the heterozygous CG genotype shows a statistically significant association in our model with reaching 90 years of age, the GG genotype points to an even higher chances of surviving beyond the age of 90. However, the effect of this might not be visible because only 14 out of 314 participants had the GG genotype, and its benefits might

have been masked by a much higher number of participants with a slightly weaker, but overall beneficial effect of the CG genotype.

The only genetic locus to reach the level of genome-wide significance ($p \leq 5 \times 10^{-8}$) in multiple GWA studies for longevity is apolipoprotein E (*APOE*) [79–81]. APOE is a protein with an important role in cholesterol transport. The *APOE* gene is polymorphic, resulting in three major isoforms of the APOE: APOE2 (ε2), APOE3 (ε3) and APOE4 (ε4) [82, 83]. The three APOE isoforms differ at the 112th and 158th residues of their primary structures, and are determined by two SNPs that cause amino acid substitutions and result in functional changes in the APOE protein: rs429358 and rs7412, respectively. APOE-ε3 (cys112, arg158) is the most common isoform of the APOE gene [84]. The carriers of this isoform have a C allele on rs7412 and T allele on rs429358. APOE-ε2 (cys112, cys158) is an isoform caused by the transition of the C allele to the T allele of rs7412, while the T allele of rs429358 remains unchanged. This mutation causes a substitution of the basic amino acid Arg158 in APOE-ε3 with the neutral amino acid Cys158 in APOE-ε2 [83], resulting in reduced APOE-ε2 receptor affinity. APOE-ε4 (arg112, arg158) isoform is characterised by the C allele of rs7412 and the C allele of rs429358. The ε4 isoform of APOE is associated with increased total cholesterol and low-density lipoprotein cholesterol [82], heart disease [85–87], Alzheimer's disease and dementia [82, 88] and other illnesses [89, 90]. Its frequency varies significantly between young adult populations. APOE-ε4 is expressed in approximately 25% of Finns, 17–20% of Danes and approximately 10% of French, Italians and Japanese. However, in all mentioned populations, the frequency of APOE-ε4 among centenarians is closer to half of these values [91]. Studies have explained this age-related distribution by showing a negative association between chances of reaching extreme longevity and the presence of ε4 [34, 92, 93]. The APOE isoforms make six possible biallelic genotypes: ε3/ε3, ε3/ε4, ε2/ε3, ε4/ε4, ε2/ε4 and ε2/ε2, which are shown here ranked from most to least common among European populations [94]. The T allele of rs7412, which is a minor allele in the Croatian oldest-old population and in all the populations indexed in 1000 Genomes database [63], has been associated with survival to the age of 90 in the studied population, both in homozygous and heterozygous form. This is in line with the findings of Deelen et al. [23], where the minor allele of rs7412 was found to have a beneficial effect on longevity. The same study found that the minor C allele of rs429358 had a deleterious effect on longevity, which is in concordance with our findings. In the Croatian oldest-old population, the rs429358 allele positively associated with survival to age 95 was the major allele T, but it was slightly below the significance level of $p \leq 0.05$. Since the T allele was associated with survival to longevity threshold age in both *APOE* SNPs, this suggests a beneficial effect of the ε2 isoform of APOE on longevity. Comparison of allele frequencies from the studied population with the European average from the gnomAD database shows that the oldest-old Croats have a lower frequency of longevity-related T allele of rs7412. This is also reflected in the lower prevalence of ε2 in the Croatian population older than 85 years compared to the data of 1038 control subjects of European origin younger than 60 years of age from a study by McKay et al. [34]. However, it is apparent that the studied Croatian population owes its longevity to a higher frequency of longevity-associated T allele of rs429358, which is confirmed by a much higher percentage of the neutral isoform ε3, and a lower percentage of the detrimental isoform ε4 (Table 3). Therefore, we can conclude that while only a small percentage of the Croatian oldest-olds benefit from the protective ε2, the majority had a good chance to reach extreme longevity by being spared from the negative influence of ε4.

The two SNPs that were significant only for survival to the age of 95 are locus rs6067484 in the *PTPN1* gene, and rs4837525 located in the *PAPPA* gene. Information on both of these loci is scarce, with only minor A allele of rs6067484 being previously associated with higher levels of total plasma cholesterol and low-density lipoprotein (LDL) cholesterol in men [95].

However, both variants were reported as potential candidates affecting longevity in a paper by Dato et al. which examined the association between SNP-SNP interactions and longevity [55]. rs6067484 is an intronic variant of the *PTPN1* gene that encodes protein tyrosine phosphatase non-receptor type 1, a suppressor of insulin signalling pathways [95]. The *PTPN1* gene is located in the q13.1-q13.2 area of chromosome 20, a region that is gained or amplified in several cancers [96]. rs6067484 of *PTPN1* was significant in interaction with rs12437963 in the *IGF1R* gene. Due to the importance of the insulin/IGF-1 pathway in ageing processes, it is not surprising that a suppressor of this pathway could be associated with longevity. This also explains why the signal for rs6067484 in the study by Dato et al. was paired with a signal for another variant involved in the same metabolic pathway [55]. In our sample, the allele contributing to survival to 95 years of age was the minor G allele, whose frequency varies from 20–30% in European and Latino populations, to only 1–4% in African populations [33, 63]. Additional connection that the *PTPN1* gene has to healthy ageing is its association with Alzheimer's disease. Studies have shown that overexpression of this gene, mediated by knockdown of miR-124, reduces synaptic failure and memory deficits, highlighting it as a promising new therapeutic target for patients with Alzheimer's disease [97, 98]. Another SNP significant in the Croatian oldest-old population is rs4837525 in the *PAPPA* gene, which also had a significant interaction with the *GHSR* gene of the insulin/IGF-1 signalling pathway in the study by Dato et al. [55]. The *PAPPA* gene encodes a zinc metalloproteinase that cleaves inhibitors of IGF, thus enhancing the activity of insulin/IGF-1 pathway [99]. This enzyme was first discovered in the plasma of pregnant women, and since its function was unknown at the time, it was named pregnancy-associated plasma protein-A [100]. A study by Bøtkjær et al. showed that another single nucleotide variant in the *PAPPA* gene caused an amino acid change (Tyr>Ser) that significantly reduced cleavage rates for one of the IGF-binding proteins [101]. The SNP observed in this study, rs4837525, is intronic and therefore does not affect the protein's catalytic activity, but could affect its expression by acting as an enhancer [63]. It is interesting that the protective effect for reaching the age of 95 years among the oldest-olds in Croatia was found in heterozygotes for this SNP, who carry one ancestral allele A and one new allele G. However, both in the studied population and in other European populations, the G allele is major, regardless of the fact that the A allele is the ancestral one.

The second part of this study focused on the calculation of genetic risk scores for reaching the ages of 90 and 95, but the scores are more aptly named genetic longevity scores (GLSs). Unweighted and weighted longevity scores were calculated for reaching the threshold ages of 90 and 95, resulting in four scores. Because longevity is a complex trait, so heavily influenced by lifestyle and environment, it is not surprising that not many studies have been conducted to quantify the chances an individual might have for longevity based on their genetic makeup. But, perhaps the most similar concept to ours is found in a study by Tesi et al., in which a polygenic risk score for predicting the odds of becoming a healthy centenarian was constructed for a population of Dutch origin using 330 genetic variants that significantly discriminated between centenarians and older adults [102]. The calculated polygenic risk score showed a statistically significant association with cognitive healthy aging and prolonged survival of a sample of 343 centenarians in good cognitive health and 2,905 population-matched controls. In the present study of the Croatian sample of the oldest-olds, observation of empirical and theoretical values of calculated longevity scores showed that no participant had a minimum or maximum theoretical value of any score. Nevertheless, participants with the lowest longevity scores fell into the category of those who died before age 90, and their mean GLS values were lower than the mean GLS values of participants who survived past any longevity age threshold. All four GLSs were significantly associated with age at death with a p value of $\leq 0.01$ in Pearson's correlation analysis, and all the results presented here support the predictive capabilities

of the calculated GLSs, which were based on only nine or five SNPs selected from a set of as few as 43 longevity variants. Namely, a multivariate linear regression analysis in which all four genetic longevity scores were compared as independent variables for their effect on age at death as a dependent phenotype, highlighted wGLS90 (p = 0.011) and uGLS95 (p = 0.001) as the most predictive scores for their threshold age for longevity. This finding was also confirmed by ROC curve analysis, with a 69% and 64.5% chance of correctly predicting survival to 90 and 95 years of age, respectively. The predictive power of the calculated genetic longevity scores is unexpectedly high, especially considering that some of the SNPs with a validated association with longevity from other studies are not even included in the models used for calculation of said scores. This might be partially due to the fact that the studied sample is pre-selected on the basis of a long life, and aged individuals in such sample have already survived the mortality selection caused by chronic diseases that arise during middle and early old ages. Therefore, the SNPs that might have had a strong negative effect earlier in life are not included when sampling a population that has already survived to an advanced old age. It is quite possible that some other loci would have been included in the models if the comparison would contrast the general adult population and the long-lived age groups.

The study of human longevity is a difficult task, as longevity phenotype is dependent on multiple other factors like individual health, genetics, environment, lifestyle differences and even chance. Because of this complexity, the approaches to studying longevity differ greatly—many studies have focused on finding the differences between long-lived individuals and non-long-lived controls [103, 104], some on studying lifespan as a quantitative variable in general populations by using survival models [105, 106], and others by observing the causal effects of specific risk factors on mortality [107]. However, all of these approaches have some shortcomings—either the choice of the right control group for a longevity GWAS, or the limited statistical power for predicting the effect of genetic variants on mortality. In a paper by Timmers et al. [108], a combination of these approaches is used to create the most comprehensible GWAS analysis of human lifespan to date, and even in that case, along with the discovery of some novel SNPs, only some of the previous findings were replicated. This further highlights the importance of undertaking longevity research with various methods, as all of them could contribute to the ever-growing pool of information on human ageing and longevity.

To our knowledge, this study is one of the first attempts to calculate genetic risk scores for longevity. The research sample consists of people who, because of their advanced age, already show the characteristics of healthy ageing. In this study, we wanted to further determine the influence of the genetic background on lifespan in this group, which was pre-selected according to age. Furthermore, we aimed to investigate whether there is a difference in genetic factors that contribute to longevity (90+ years) and those that could play a role in reaching extreme longevity (95+ years). The sex distribution of the studied population leans to the female side, with 74.8% percent of female participants, which is in line with the structure of the general population of Croatia for that age group, where 74.5% of people over 85 are women [109]. The main limitation of this study was the small number of SNPs available for analysis, which was compensated by the selection of genetic variants with a strong previously reported association with longevity and a role in various cellular pathways associated with the ageing process. The number of participants was also relatively small, but there was no pooling of data from several studies of independent populations that could weaken estimates of genetic associations due to the different environmental effects or genetic backgrounds [110], which were quite homogenous in this sample.

To summarise, this study indicates which of the previously reported SNPs also correlates with longevity in the Croatian population, a European population whose genetic data are still underrepresented in the available literature. Regardless of the various factors that may

influence age at death, including stochastic events, a selected set of longevity-associated SNPs explains a noteworthy 20% of the variance for survival to age 90 in a Croatian sample of the oldest-old individuals (85+ years). Of the analysed SNPs, rs16847897 in the *TERC* gene and rs2267723 in the *GHRHR* gene were most significantly associated with longevity in the model for survival to the age of 90 with a $p \leq 0.01$, while the set of genes affecting extreme longevity was quite different and with somewhat weaker associations ($0.01. < p < 0.05$). This study also provides unweighted and weighted genetic risk scores for predicting survival to the threshold ages of longevity (90) and extreme longevity (95 years), and while all four calculated scores were significantly correlated with longevity, wGLS90 had the highest predictive accuracy.

## Supporting information

**S1 Table. Information about the selected SNPs: Rs code, nearest gene, chromosome position, references for literature sources in which association with longevity was reported; along with data that refers to the studied Croatian population: Alleles (major/minor), minor allele frequencies (MAF), genotyping success rate, Hardy-Weinberg equilibrium (HWE).**
(XLSX)

**S2 Table. Results of univariate binary logistic regression analysis for cut-off ages at death.**
(XLSX)

**S3 Table. Means, standard deviations and differences in genetic longevity scores between sexes calculated using Student's t-test.**
(XLSX)

**S1 Fig. Absolute distribution of the four genetic longevity scores median values (equal proportion of participants having genetic longevity scores above and below the median) by three age-at-death groups (<90.00 years, 90.00–94.99 years, and 95.00+ years).** A) uGLS90, B) uGLS95, C) wGLS90, D) wGLS95.
(TIF)

**S2 Fig. Relative distribution of age-at-death groups by median genetic longevity score (equal proportion of participants with genetic longevity scores above and below the median).** A) wGLS90, B) wGLS95.
(TIF)

## Author Contributions

**Conceptualization:** Nina Smolej Narančić, Tatjana Škarić-Jurić.

**Data curation:** Tanja Ćorić, Branko Kolarić, Nina Smolej Narančić, Tatjana Škarić-Jurić.

**Formal analysis:** Maja Šetinc, Željka Celinšćak, Tatjana Škarić-Jurić.

**Funding acquisition:** Tatjana Škarić-Jurić.

**Investigation:** Matea Zajc Petranović, Marijana Peričić Salihović, Nina Smolej Narančić, Tatjana Škarić-Jurić.

**Methodology:** Maja Šetinc, Željka Celinšćak, Tatjana Škarić-Jurić.

**Project administration:** Tatjana Škarić-Jurić.

**Resources:** Luka Bočkor.

**Supervision:** Luka Bočkor, Branko Kolarić.

**Validation:** Maja Šetinc, Željka Celinšćak, Luka Bočkor, Tanja Ćorić.

**Visualization:** Maja Šetinc.

**Writing – original draft:** Maja Šetinc.

**Writing – review & editing:** Maja Šetinc, Željka Celinšćak, Luka Bočkor, Tanja Ćorić, Branko Kolarić, Anita Stojanović Marković, Matea Zajc Petranović, Marijana Peričić Salihović, Nina Smolej Narančić, Tatjana Škarić-Jurić.

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
