## [Decision Letter · Decision Letter 0]

15 Nov 2022

PONE-D-22-29477Genetic scores for predicting longevity in the Croatian oldest-old populationPLOS ONE

Dear Dr. Šetinc,

Thank you for submitting your manuscript to PLOS ONE. After careful consideration, we feel that it has merit but does not fully meet PLOS ONE’s publication criteria as it currently stands. Therefore, we invite you to submit a revised version of the manuscript that addresses the points raised during the review process. In particular, the reviewer requests clarification on interpretation of results.

We look forward to receiving your revised manuscript.

Kind regards,

David M. Ojcius

Academic Editor

PLOS ONE

Reviewers' comments:

Reviewer's Responses to Questions

**Comments to the Author**

1. Is the manuscript technically sound, and do the data support the conclusions?

Reviewer #1: Partly

2. Has the statistical analysis been performed appropriately and rigorously? 

Reviewer #1: No

3. Have the authors made all data underlying the findings in their manuscript fully available?

Reviewer #1: Yes

4. Is the manuscript presented in an intelligible fashion and written in standard English?

Reviewer #1: Yes

5. Review Comments to the Author

Reviewer #1: Šetinc and colleagues present results from a study of long-lived Croatian’s using targeted genotyping methods. Of the 43 variants genotyped – each with strong a priori evidence from the literature – two (2) were associated with survival to age 90+ in the multivariate model with p<0.01. In the univariate analyses, only one variant (rs2267723, GHRHR) was associated with p<0.01.

Major comments:

1. The selection of the 43 variants includes many key longevity loci from the field (APOE, SH2B3, CDKN2A, etc.) but is missing some key references. In particular the Timmers et al. 2019 meta-analysis of 500,000 people for parents lifespan (https://pubmed.ncbi.nlm.nih.gov/30642433) which identified several variants not included in the targeted genotyping panel. I am of course not asking you to re-do genotyping, but please acknowledge/justify this omission.

2. Table 1 (and 2) present results from a multivariate model where the genotype specification is chosen based on the best model fit parameters. This has resulted in some surprising results, for example participants heterozygous for rs1042522 located in the TP53 gene (i.e., the CG genotype group) are reported to have 1.7x likelihood of reaching 90 years, compared to the common genotype groups (CC + GG combined). Whilst there is a lot in the Discussion on p53 role in ageing/longevity the authors do not comments on why the heterozygotes only would experience lifespan benefit – a result in contract to the previous literature (which saw an additive effect with increasing G alleles).

2.1. I suspect that these borderline associations (p=0.04) should not be read into much, given the sample size and therefore power to detect the published effect sizes. The authors report substantially larger effects than seen in prior literature, after a running multiple iterations of the model to get the best prediction. I am not sure on the biological validity of including variants coded as HET vs HOM+HOM – normally in genetic epidemiology this format would be used to test for deviation from an additive model (i.e., justifying whether to use a recessive or dominant model).

3. I personally put more belief in the univariate results available in Supplementary Table 2, where 5 SNPs were associated (p<0.05) with survival to 90+ (and one for 95+). I suggest more emphasis it put on these results in the paper as this seems better “replication” of the published effect of these SNPs. Whilst the p-values are quite borderline, each SNP has a priori evidence meaning multiple testing correction would be too conservative, and using p<0.05 is appropriate, given the analysis.

4. I am actually surprised that the polygenic score is predictive of 5-year survival, given all the participants by definition reached 85 years to be eligible for the study, and the main outcome was survival to 90 years. Many of the studied “longevity” variants are actually more “disease risk” variants likely affecting mortality risk at ages below 85. There even reaching 85+ is a selected group, and this may explain why some “longevity” variants from the literature do not appear to have an effect in your study (i.e., the people who experienced the negative effects were not included?). Please discuss this point in the manuscript (what effect the participant selection had on power etc).

Minor comments:

1. Please include sample size in the abstract (N in study, % reached 90+/95+, etc)

6. PLOS authors have the option to publish the peer review history of their article (what does this mean?). If published, this will include your full peer review and any attached files.

Reviewer #1: No

---

## [Author Response · Author response to Decision Letter 0]

17 Dec 2022

Response to Editor:

We would like to thank Professor Ojcius for all the helpful advice he has given us, and for the opportunity to improve our manuscript and submit it again.

In his letter, he brought to our attention three issues - proper file naming that meets PLOS ONE's style requirements, the need to deposit our study’s minimal underlying data set as either Supporting Information files or to a public repository, and the mention of one subset of data as 'not shown'. We have done our best to resolve these issues, and our new manuscript should now be formatted according to PLOS ONE's requirements. Also, the dataset used in our study is now publicly available in the public repository Zenodo, and is accessible via this link: https://zenodo.org/record/7421684. Lastly, the data that has been referred to as 'not shown' is now available as Supporting Information in S3 Table.

In this resubmission, we have included a new cover letter, a rebuttal letter for the Reviewer, as well as marked-up and unmarked versions of our new manuscript. We hope that the improvements made have brought our work closer to PLOS ONE’s publication criteria, and once again want to express our thanks to Prof Ojcius for his editorial advice.

Response to Reviewer:

We would like to sincerely thank the Reviewer for their constructive criticism and advice which have aided us greatly in improving our work. Along with addressing the Reviewer's comments here in short, our full response can be found in the 'Response to reviewers' file.

The Reviewer pointed out the omission of a paper by Timmers et al. (2019) from our bibliography, and the key SNPs found in that study from the pool of SNPs that we have chosen to genotype in our sample. Unfortunately, we were not aware of the paper by Timmers and colleagues at the time we did the selection of SNPs from the literature in the beginning of 2019, but we have now included the mention of this paper and its methodology in our manuscript.

The Reviewer asked us to comment on why the heterozygotes of TP53 rs1042522 would experience lifespan benefit in the model for survival to 90 years of age, when previous literature points to beneficial effect of homozygous GG. We have elaborated on the different benefits of alleles C and G in relation to cancer risk and longevity, and have proposed a possible explanation for the effect we see in our study. We have also explained how the population distribution of this allele in our relatively small sample could have skewed the estimated effects of these alleles.

We have put more emphasis on the results of our univariate analyses, as the Reviewer suggested, and explained why one of the SNPs that was statistically significant on the univariate level had to be excluded from multivariate models.

We have also explained how the selection of a long-lived sample for our study could have excluded certain variants that have a stronger effect in earlier stages of life, possibly affecting the power of our study.

Lastly, per Reviewer’s suggestion, we have included sample sizes of different survival groups in our abstract.

---

## [Editor Report · Decision Letter 1]

19 Dec 2022

Genetic scores for predicting longevity in the Croatian oldest-old population

PONE-D-22-29477R1

Dear Dr. Šetinc,

We’re pleased to inform you that your manuscript has been judged scientifically suitable for publication and will be formally accepted for publication once it meets all outstanding technical requirements.

Kind regards,

David M. Ojcius

Academic Editor

PLOS ONE